# Social norms and maternal health information-seeking behavior among adolescent girls: A qualitative study in a slum of Bangladesh

Md. Ashraful Haque[1]*, Rabeena Sultana Ananna[2], Nayeem Hasan[3], Mst. Farhana Aktar[4], A. F. M. Zakaria[1]

1 Department of Anthropology, Shahjalal University of Science and Technology, Sylhet, Bangladesh, 2 BRAC's Institute for Governance and Development (BIGD), Dhaka, Bangladesh, 3 CARE Bangladesh, Dhaka, Bangladesh, 4 Jahangirnagar University, Savar, Dhaka, Bangladesh

* ashraful.sustedu.anp@gmail.com

## Abstract

Adolescent girls of reproductive age who actively seek information on maternal health often tend to have better health-seeking behaviors and maternal health outcomes. Due to scant research on reproductive aged adolescent girls' maternal health information seeking behavior in slum, in connection with social norms, we aimed for this particular study. Adopting an explorative qualitative research approach, we collected data from purposively selected married and unmarried adolescent girls aged 15–19 of different occupation by implying 12 in-depth interviews (IDIs), 2 focus group discussions (FGDs) with the same categories employed for IDIs, and 2 key informant interviews (KIIs) with a traditional birth attendant and a drug seller. Furthermore, the data were subjected to thematic analysis. Care's Social Norms Analysis Plot (SNAP) framework was undertaken as an interpretative tool for data that was emerging rather than serving as the foundation for the study's conduct and design. Thematic analysis was followed to analyze primary data. Findings show that most girls rely on maternal health-related information from unverified sources, including family members, traditional birth attendants, and drug sellers, which increases health risks. The majority reported that adolescent girls need professional healthcare providers in their area who would work according to their work schedule as most of the girls are engaged in income-generating work for about 9–11 hours, and the scope of work (daily wagers) hardly supports 'leave with pay'. Therefore, there is a critical need for professional healthcare services tailored to the girls' work schedules. Social norms and stigma further restrict access to reliable health information, especially for unmarried girls. Socioeconomic disparities also shape health-seeking behaviors, with wealthier adolescents having greater access to formal healthcare services. Addressing these barriers is crucial for improving maternal health outcomes. The results might be useful for informed policy formulation and program design to ensure better health outcomes for marginalized adolescents.

**Data Availability Statement:** All data are in the manuscript and supporting information files.

**Funding:** The author(s) received no specific funding for this work.

**Competing interests:** The authors have declared that no competing interests exist.

**Abbreviations:** BDHS, Bangladesh Demography and Health Survey; CHW, Community Health Worker; FGD, Focus Group Discussion; HISB, Health Information Seeking Behavior; IDI, In-depth Interview; KII, Key Informant Interview; TBA, Traditional Birth Attendant; WHO, World Health Organization.

## Introduction

Maternal health is a worldwide issue, even though it was reduced by 34% from 451,000 in the year 2000 to 287,000 deaths in 2020. This is a particular problem for developing countries where 95% of maternal deaths occur [1]. In Bangladesh, reducing maternal mortality is a national priority, supported by policies like the Bangladesh National Health Policy 2011 [2]; as long as with the Bangladesh National Strategy for Maternal Health 2019–2030 [3]. The issues of maternal health are in the attempts to reduce all the factors that negatively affect the physical condition and social position of women and mothers.

Bangladesh has one of the highest adolescent fertility rates in the Asia Pacific, with 128 births per 1000 girls aged 15 to 19 [4]. Adolescent girls are married off 3–4 years earlier than the legal marriage age of 18. Married adolescent girls go through unavoidable social and family pressure to get pregnant soon after marriage as proof of their fertility [5]. Their incapacity to obtain family planning (FP) and reproductive health (RH) services encourages early childbirth and marriage [6]. In contrast to rural women, women living in urban regions are more likely to have made four or more antenatal visits (59% vs. 43%) [7]. Again, the health-seeking behavior of people living in cities varies. Evidence illustrates that 20% of teenage mothers gave birth unintendedly [5], and unwanted pregnancies are more than twice as common among married adolescent girls in Bangladesh's slums as in non-slum areas [6]. Previous study demonstrates that Bangladeshi slum had a maternal mortality rate of 294 deaths per 1000,000 live births and a neonatal mortality rate of 43 deaths per 1,000 live births [8].

Adolescent girls in Bangladesh have considerable obstacles when attempting to get maternal health information and treatments, mostly because of their dependence on unofficial health-care professionals and older family members. Despite not having a medical degree, traditional birth attendants, neighborhood drug sellers, and older female relatives—like mothers-in-law—are frequently the main providers of guidance on maternal health. Adolescent moms may engage in unsafe behaviors and suffer negative health consequences because of their dependence on unreliable sources [9]. Furthermore, teenagers' work schedules frequently prevent them from accessing healthcare facilities; this is especially true for those in physically demanding professions with little room for flexibility or time off [10]. Adolescent girls, especially those who are married, tend to rely on the guidance of older family members, which might limit their access to official maternity healthcare. This is where social and cultural norms come into play [11]. Improving maternal health outcomes among Bangladeshi adolescents requires addressing these obstacles.

Access to timely and relevant maternal health information is crucial for informed decision-making and reducing maternal morbidity and mortality. Accurate maternal health information promotes positive health behaviors and better maternal health outcomes [12]. Delays in seeking appropriate care during pregnancy significantly increase maternal risks. While some studies have addressed maternal and neonatal mortality in slums, there is a notable gap in research on the health information-seeking behavior of adolescent girls in Bangladesh's urban slums. This is critical, given that slums host over 5.7 million people, approximately 3.8% of the national population, most of whom are migrants seeking better economic opportunities [13]. Addressing the maternal health information needs of this marginalized group is essential for improving maternal health outcomes in Bangladesh.

Care's Social Norms Analysis Plot (SNAP) Framework was developed to measure the nature of specific social norms and their influence and offers a useful framework to examine the initial reactions to a social norm's activity. Empirical expectations, normative expectations, sanctions, sensitivity to sanctions, and exceptions are five key constructs to understand one's act or decision-making [14]. Although studies exist on overall unexpected pregnancy and

contraceptive use among married adolescent girls in Bangladesh, data on particularly vulnerable groups (as our study examines both married and unmarried adolescents from an urban slum) of adolescents is scant. In addition, to our best knowledge, there is no research done yet to understand the maternal health care information seeking in social norm perspective. Therefore, we aim to explore the maternal health information-seeking behavior of adolescent girls of reproductive age in a slum through the lens of social norms. The primary goals of this study were to identify adolescent girls' maternal health information needs during their reproductive years and to learn about how social norms working as the obstacles they face when seeking information related to maternal health.

## Methodology

### Study design

The research team for the study consisted of five members (three males- MAH, NH, and AFMZ; and two females- RSA, MFA). The only geographer on the team is MFA. The rest of the researchers graduated from Anthropology. We undertook an exploratory qualitative method, a widely used methodological approach, especially within practice disciplines [15], to understand maternal health information-seeking behavior among adolescent girls of reproductive age in a slum of Dhaka, Bangladesh. We used the term- adolescent girls of reproductive age by considering the mostly known definition of adolescents, who are from 10–19 years, and the reproductive age of a woman, which is 15–49 years, according to WHO [16, 17, 21]. In such a sense, adolescent girls of reproductive age refer to girls who are between 15 and 19 years old. This approach was selected for this study due to its nature of nature of investigating research objectives by exploratory questions [18]. The lack of in-depth research on maternal healthcare access among this age group living in slums led us to a need for an in-depth exploratory study. For this, we conducted 12 in-depth interviews (IDIs), 2 focus group discussions (FGDs) and 2 key informant interviews (KIIs) (Table 1), to provide a data source for triangulation. A total of 26 participants were recruited, whereas 34 were approached. The two reasons for rejecting interviews were time and unwillingness. IDI participants provided subjective viewpoints and FGD participants helped us to understand the community perspective. Participants of KII facilitated to get objective viewpoints of the data collected by other methods.

We developed separate guidelines for IDI, KII and FGD participants. MAH and RSA prepared the draft guideline. Then, it was reviewed and furnished by the other authors. As all the guides had similarities in the question pattern [18], we did a pilot test of the IDI guideline only in a different field. The Consolidated criteria for Reporting Qualitative research (COREQ) guided the reporting of the study (S1 Table) [19, 20].

**Table 1. Participant size and methods of study.**

| Method | Participant Group/Type | Total |
|---|---|---|
| **FGD (2)** | Married (Adolescent girls) | 6 |
| | Unmarried (Adolescent girls) | 6 |
| **IDI (12)** | Married (Adolescent girls) | 6 |
| | Unmarried (Adolescent girls) | 6 |
| **KII (2)** | TBA | 1 |
| | Drug seller | 1 |
| **Total 26** | | |

## Study setting

A number of slums are seen in Bangladesh, especially in Dhaka City. Over 45 million people live in urban regions in Bangladesh, where the share of the population living in urban settings has climbed from 5% to 28% in the last 40 years [21]. Unprepared cities with inadequate infrastructure find it difficult to handle this constant flood of migrants. New urbanites have difficulties in the so-called arrival city when they live in slums or on the streets [22]. The number of slums and urban poverty in Bangladeshi cities like Dhaka is overwhelming. The already challenging living circumstances of the impoverished in cities are made worse by extremely high population densities and failing municipal infrastructure. The study was done on Sattola, a slum of Dhaka city, Bangladesh because of its highly dense population and urban slums, which are well-known for the highest rate of teenage mother. From our observation, the area is shoddy, with below-standard houses built at less expense; this results in inadequate sanitation and deficiencies in access to basic services such as health insurance. The people of Sattola are not only battling with high rates of adolescent pregnancy, but they are also subject to a paramount of material constraints. No particular NGO has been found working particularly on adolescent maternal health care during fieldwork. However, there are some treatment facilities which is available at the International Centre for Diarrheal Disease Research, Bangladesh, and Universal Medical College and Hospital, which provide some diagnosis services related to maternal health. These medical facilities are situated within a 2–3 kilometer radius of most of the households. To maximize comprehension and enable correct response documentation, the IDI, FGD and KII guides with vignettes were tested, revised, and collaboratively translated into Bengali and back into English.

## Sampling and recruitment

The study used a purposive sampling method to recruit study participants at their convenience due to the variety of occupational engagements of the study population [23]. Purposively, our data collectors, MFA (co-author), SS, FS, and AS, visited the slum to find potential participants. To ensure confidentiality and consider the sensitivity of the research topic, when discussing delicate subjects, participants were assigned with same-sex data collectors. The data collectors were trained anthropologists and geographers. The field researchers received three days of training from the lead investigator (MAH, AFMZ), with a focus on sensitive research involving human subjects and notably adolescent subjects conducted in an ethical manner. The inclusion criteria for IDI were: a. availability, b. age (15–19), c. female adolescent (married and unmarried), and d. willingness.

## Data collection procedure

It took about one month to complete data collection. We collected data from the field from the 10th of November 2022 to the 15th of January 2023. The field researchers shared about themselves first. Then, they explain the reason for their arrival to the field, study objective, and goal to the participants. As most of the participants were engaged in diverse types of professions, we mostly had to gather data on their weekends or off days (Tables 2 and 3). After having a preliminary discussion and validating all the requirements, the participants were selected. All the interviews lasted from 50 to 75 minutes and were taken in their recommended place and time. The field researchers ensured that the interviews were conducted in a quiet place. However, in the context of the slum, it was quite difficult as they lived in a one-room house. Therefore, we need to plan according to the participant's given time. Moreover, before starting the interview, oral consent was taken, and permission for audio recording was requested. Interviewers maintained field notes during every interview and discussion.

**Table 2. Demographic profile of the IDI participants.**

| Category | ID | Age | Age at Marriage | Education (passed) | Occupation | Average Working Hour | Off Day |
|---|---|---|---|---|---|---|---|
| **Unmarried (Adolescent girls)** | 1 | 15 | N/A | G-2 | Garment Worker | Not fixed | 1 |
| | 2 | 17 | N/A | G-8 | Tailor at home | 10 | 1 |
| | 3 | 18 | N/A | G-5 | Garment Worker | 10 | 1 |
| | 4 | 18 | N/A | G-9 | Garment Worker | 10 | 1 |
| | 5 | 17 | N/A | G-5 | Garment Worker | 10 | 1 |
| | 6 | 17 | N/A | G-7 | Day Laborer | Not fixed | N/A |
| **Married (Adolescent girls)** | 7 | 16 | 16 | G-8 | Garment Worker | 9.5 | 1 |
| | 8 | 17 | 17 | G-5 | Tailor at home | Not fixed | 1 |
| | 9 | 18 | 17 | G-5 | Garment Worker | 10 | 1 |
| | 10 | 15 | 15 | G-5 | Garment Worker | 9.5 | 1 |
| | 11 | 19 | 17 | G-3 | Garment Worker | 9 | 1 |
| | 12 | 18 | 17 | G-10 | Garment Worker | 10 | 1 |

Before commencing the fieldwork, we intended to interview at least eight people from each group (married and unmarried). However, after completing five interviews, we hit data saturation [23]. Then, we conducted further interviews with each group to be more certain. Among two KII participants, one was a local traditional birth attendant (TBA), and the other was a drug seller by profession. TBA was recruited as the majority of the study participants felt comfortable and used to visit her. On the other hand, drug seller is the key person in the slum to take medicine. Participants consider them as doctors and take medicine from them without a prescription, rather than just telling health problems. Despite UAGs' and MAGs' access to *Kabiraj* (faith healer), we could not interview him as he was outside of the slum due to personal reasons. Furthermore, we approached formal health service providers whose facility was closer to the slum; however, as he reported that he received any adolescent girl from the studied slum as a patient, we did not consider them as KII.

## Data analysis

All the qualitative data was organized using Microsoft Excel, and six phase thematic analysis was performed [24]. Instead of relying on a pre-existing category, categories emerged inductively from the data. Data related to the study's main topics and emergent themes from the

**Table 3. Demographic profile of the FGD participants.**

| No. | Category | ID | Age | Age at Marriage | Education (passed) | Occupation | Average Working Hour | Off Day |
|---|---|---|---|---|---|---|---|---|
| FGD 1 | **Unmarried Adolescent Girls (UAGs)** | 13 | 16 | N/A | G-9 | Garment Worker | Not fixed | 1 |
| | | 14 | 17 | N/A | G-5 | Garment Worker | 10 | 1 |
| | | 15 | 18 | N/A | G-7 | Garment Worker | 10 | 1 |
| | | 16 | 16 | N/A | G-5 | Day Laborer | Not fixed | N/A |
| | | 17 | 15 | N/A | G-5 | Tailor at home | 10 | 1 |
| | | 18 | 15 | N/A | G-5 | Garment Worker | 9.5 | 1 |
| FGD 2 | **Married Adolescent girls (MAGs)** | 19 | 16 | 14 | G-5 | Garment Worker | 9.5 | 1 |
| | | 20 | 18 | 16 | G-7 | Garment Worker | 10 | 1 |
| | | 21 | 17 | 16 | G-5 | Garment Worker | 10 | 1 |
| | | 22 | 19 | 15 | G-10 | Tailor at home | Not fixed | 1 |
| | | 23 | 19 | 17 | G-4 | Garment Worker | 9 | 1 |
| | | 24 | 18 | 16 | G-3 | Garment Worker | 10 | 1 |

first review were collected by codes. The coders (MAH, MFA, RSA, NH) defined codes that seemed meaningful in an iterative method. Despite the fact that the codes came from the data, the coders admit that their experiences and background impacted the codes they defined. Representative quotes from the transcript were coded after the codes had been defined and identified.

After codes were examined, overlapping codes were further categorized. Following that, classifications representing a degree of pattern in reaction or meaning were used to create themes. Themes that mirrored experiences with wellness and decision-making related to maternal health care were determined to be salient. The methodology for data analysis was a theory-informing inductive data analysis research design, which means that the theory—in this case, Care's Social Norms Analysis Plot (SNAP) Framework—was applied as interpretative tool for data that was emerging rather than serving as the foundation for the study's conduct and design [14].

## Ethical consideration

The study followed ethical issues with high priority as it ensured that there would be no harm to the participants and maintained confidentiality, as we promised them before the interview [18]. All of the participants gave their oral consent. However, for participants under 18, we took assent from their parents or guardians. One of the team members (MAH) was appointed to monitor these ethical issues. He took a small session on ethical issues during the training for data collection and got ensured about the application of ethical consideration after the daily de-briefing session during fieldwork time. The research team members discussed all these issues in the daily de-briefing session. As most of the participants were working girls, most of the interviews were done on their off days and after their office hours. Before starting the interviews, all the relevant issues were discussed, and questions from the study participants were answered. It is planned that after publication of the manuscript, all the data will be destroyed, as promised to the study participants.

## Ethical clearance

This study received ethical approval from the Ethical Approval Committee of the Department of Anthropology, Shahjalal University of Science and Technology, Sylhet, Bangladesh. The approval number for this study is ANP_AR_2022:10.

## Result

The result section is divided into several sub-sections. Our primary focus is to understand maternal health information-seeking behavior. We believe that it would be better understood if we first looked into the need, then the sources of information, and finally, the barriers or, more specifically, the influences of social norms.

## Maternal health information needs for adolescent girls

We collected data on maternal health information needs using the IDI, KII, and FGD methods. Findings demonstrate that MAGs and UAGs of reproductive age in the slum have the greatest need for information on early indicators of pregnancy, risk signs of pregnancy, abortion, obstetric fistula, medication, and an accessible contact person (Table 4). A critical examination of the data reveals that both practical and strategic gender needs influence the health-seeking behaviors of adolescent girls.

**Table 4. Emerged themes and sub-themes.**

| Sl. | Theme | Definition | Sub-theme |
|---|---|---|---|
| 1 | **Maternal health information needs** | The maternal health information that a married or unmarried adolescent need | Early indicators of pregnancy |
| | | | Risk signs of pregnancy |
| | | | Abortion |
| | | | Obstetric fistula |
| | | | Medication |
| | | | Accessible contact person |
| 2 | **Sources of Information** | UAG's and MAG's sources of information | Trusted sources |
| 3 | **Barriers** | All the contextual factors that prevent an adolescent girl aged 15–19 to get maternal health information, not social norms | Contextual factors |
| | | | Social norms |

Practical gender needs refer to the immediate necessities such as the knowledge of early pregnancy indicators and risk signs, while strategic gender needs pertain to addressing broader issues like autonomy and empowerment.

Findings from the study illustrate that most of the adolescent girls in the slum need a variety of information related to maternal health. This information could help them decide whether to conceive and the treatment process during and after pregnancy, at least by knowing their actual health condition. Adolescents hardly know the actual early signs of pregnancy, which results in unwanted and unintended pregnancies. At present, they consider themselves pregnant when their period has been missing for a long time. Neither do they consult with a doctor, nor do they take any pregnancy tests. Whenever the bump is clearly visible, it is too late to make early decisions. As a result, they also must face domestic and intimate partner violence, such as verbal abuse, mistrust, and beatings, because they conceive a baby without their in-law's concern. One married IDI participant pointed out,

> "*I did not do anything wrong. But, after being pregnant, when the family members notified me based on the symptoms, it was too late. My husband and I were not ready to have a second child as our economic situation was not solvent. Because of this, my husband did not talk to me for more than seven months.*" (IDI 8, married adolescent girl, age 17)

Majority of the participants reported that the husband and the in-laws are the key decision makers of seeking maternal care information, and then deciding about any health decision, such as visiting doctors, checking up, following up, timing of carrying baby and abortion.

Adolescent girls only know that if their menstruation cycle is interrupted, it is a symptom of pregnancy. However, most of them hardly know the next steps. They do not know in which phase of their pregnancy they should seek health services and receive suggestions. Also, they hardly have any knowledge about the attention that they require during the pregnancy period. Abortion is very common in the slum. Both the married and unmarried girls were asked to share their knowledge and stories of abortion. However, only the married adolescent girls reported their abortion stories to us. Among the 12 married IDI participants (Table 3), many reported that they had aborted their unwanted child. Most interestingly, among the five cases, four were aborted while pregnant with a second child.

One married participant said,

> "*My husband and I knew that aborting a child is a sin. However, what would we do? We did not want that child at that time. Unwillingly, that came into my womb. Already, I've had a*

*child for one year! How could we care for more than one child? We neither had enough money nor could we spend time on them, as I could not join my job at that time.*" (IDI 8, married adolescent girl, age 17)

Another married girl reported,

"*My husband took me to a clinic. He planned to abort our second baby without consulting me. I was too shocked to know that suddenly. He also forced me not to share this matter with anyone. When I refused his proposal, he threatened to divorce me. He killed my baby (sobbing)!*" (IDI 9, married adolescent girl, age 18)

These accounts highlight how strategic gender needs, such as reproductive autonomy and the ability to make informed decisions about family planning, are unmet due to socio-economic constraints and patriarchal pressures.

Though KII participants clarified that they know of some cases of abortion by UAGs, no unmarried IDI participant was keen to share those pieces of information with us. The major reasons of not reporting about abortion by the UAGs were existing social stigma. To illustrate, if people of the slum come to know that any particular UAG has aborted child, she would never be able to get married in their slum. Moreover, her and her family's dignity would be diminished.

"*In the slum, if any UAG is identified that she got pregnant, her life would be finished. If she has any younger sister, she would be treated badly and harassed by young boys of the slum. Overall, her family would be losing its dignity, fully.*" (KII 2, drug seller, age 39)

Several participants reported a lot of misconceptions regarding obstetric fistula. Girls need to know the causes, potential therapies, and preventative measures for obstetric fistula. Majority emphasized how crucial it is for them to have knowledge about fistulas since it will increase their awareness of the issue and prepare them for what to do if they develop one.

"*Isn't it a periodical complication? We don't know more. Is this risky for health?*" (FGD 2, married adolescent girls)

Both MAGs and UAGs were keen to know whether the use of traditional medicine and visiting traditional birth attendants and *Kabiraj* (faith healers) was reliable or not. They discussed that their ancestors had used these medicines made of herbs and visited the traditional birth attendants and *Kabiraj* (faith healers) over the years. Therefore, they also follow them. However, they were confused about the effectiveness of those treatments.

"*Poor people like us are used to getting treatment from the local Kabiraj and traditional birth attendants. The birth attendants see the patients according to their past experiences. On the other hand, Kabiraj visits us, listens to our problems, and gives us some medicine made of various types of herbs. We receive these treatments. Some of us get a quick recovery, and some do not get well. Then, we take those patients to the nearby hospital.*" (FGD 1, unmarried adolescent girls)

The key informant, who is a male drug seller, reported that the adolescent girls hardly know the family planning method. In most cases, they do not know how to deal with sexual intercourse from the very beginning. As a result, they do what their intimate partners (husbands in

the case of married girls and sometimes boyfriends in the case of unmarried girls) want them to do. Because of this lack of knowledge, their right to give consent in the decision-making process is also hampered.

"*The adolescent girls who are married at an early age cannot give their opinion about whether they want to be pregnant or not just because of their lack of knowledge regarding family planning methods. As a result, they become pregnant, realizing that the upcoming baby has become a burden for them.* (KII 2, drug seller, age 39)

On the other hand, the female key informant, a health assistant by profession, argued that the ultimate reason for the lack of knowledge is not always from the female or adolescent's side. As the fathers or husbands are the key decision-makers in the community, the females cannot express their opinion regarding any health service availabilities or family planning decisions, even though they know the information.

"*In a male dominant society, it is quite tough for older females to establish their decision or choice. In such a condition, how can one imagine that the adolescents, who are counted as children in our community, would be able to know and comply with the family planning methods and others in their lives? That's funny. . . . . . .*" (KII 1, TBA, age 52)

The KIIs also added that adolescents barely know the importance of exercise during their pregnancy period. They said that exercise can keep the baby in the right position. In addition, they do not know concepts like kangaroo mother care, breastfeeding, and communicable diseases, which are very necessary for mothers to know so that they can give birth to healthy babies.

This is also clear from the IDI, FGD, and KII data that adolescent girls have very little knowledge about the basic information regarding maternal health. In such conditions, when we asked the girls from both groups whether they knew about post-partum and post-natal psychosis, all the participants replied negatively. As they hardly knew the concepts, we gave them some examples. However, the results appeared to be negative. As a result, they said,

"*We do not know about these terms. However, after listening to you, we think that adolescents from slums need knowledge of such information more than any other age group.*" (FGD 2, married adolescent girls)

It is mentionable, findings show that the urge to seek maternal health information among adolescents can sometimes be triggered by unexpected incidents like maternal and neo-natal deaths. For instance, in 2021, three married adolescent girls died during and after their pregnancy. These incidents create fear among adolescent girls. In their opinion,

"*In our slum, at least three adolescent girls died in the last year (in 2021). Among the cases, two of them died because none of them, including their family members, sought proper treatment from any formal sources. Those deaths made the slum girls more careful during pregnancy.*" (FGD 1, unmarried adolescent girls)

## Sources of health information

The findings show that married adolescent girls (MAGs) mostly rely on informal sources such as family members, traditional birth attendants, and neighborhood health care professionals

**Table 5. Sources of maternal health information for married and unmarried adolescent girls.**

| Source of Information | Married Adolescent Girls (MAGs) | Unmarried Adolescent Girls (UAGs) |
|---|---|---|
| **Informal Sources** | | |
| *Family Members* | Receive guidance from older female relatives (mothers, mothers-in-law, aunts). | Rarely discussed. Family members do not share information until after marriage. |
| *Traditional Birth Attendants* | Frequently consulted for accessible and affordable maternal care, despite informal expertise. | Rarely consulted, as maternal health is not considered relevant pre-marriage. |
| *Peers/Neighbors* | Share informal experiences about pregnancy and maternal health. | They discuss these issues with peers, not neighbors. |
| *Local Informal Health Providers (Kabiraj)* | Often consulted for low-cost advice, though their practices are not evidence-based. | UAGs do not generally seek advice from local health providers on maternal health. |
| **Formal Sources** | | |
| *Health Clinics/NGOs* | Limited access due to poverty, distance, and work-related time constraints. Some NGOs provide outreach, but not consistently. | UAGs do not seek formal health services, believing maternal health is irrelevant until marriage. |
| *Educational Materials (Print/Media)* | Limited access to media campaigns or printed materials, constrained by low literacy and time. | They learn such information form posters and advertisement |
| *Formal Healthcare Providers* | Rarely accessed due to financial and time barriers, with preference for informal care. | UAGs do not visit formal healthcare providers for maternal health information. |

(Table 5). Their minimal use of formal healthcare is a result of time and financial restrictions. Mother and Sister-in-law, in particular, are the most reliable sources of knowledge for them. Conversely, unmarried adolescent girls (UAGs) rarely seek information on maternal health, either from informal or formal sources as social norms and taboos surrounding discussions of maternal health before marriage significantly limit their engagement in these topics. UAGs' most trusted sources of information are peers and advertisements. In the parts that follow, the rationale for the several carefully chosen and reliable sources is covered in detail.

## Barriers encountered by adolescent girls while seeking maternal health information

From the findings, we found that poverty, lack of knowledge, education, time constraints, and existing social norms are the foremost barriers for adolescent girls regarding maternal health information seeking. To understand the existing barriers of MAGs and UAGs, we divided the theme barriers into two sub-themes: 1. Contextual factors and 2. Social Norms (Fig 1).

**Contextual factors.** One of the major barriers for adolescent girls seeking maternal health information is their office schedule. From Tables 2 and 3, we can see that most adolescent girls are directly connected to income-generating activities, especially in the garment sector. The study participants reported that most of the garments provide only one off-day per week- Friday. As a result, the health assistants or health associates, who visit door-to-door, hardly reach the adolescent girls, and vice versa.

The demographic profiles reveal the intersectionality of these adolescent girls' identities, including their age, marital status, occupation, and educational background. These factors compound their vulnerabilities and influence their access to maternal health information. The study highlights the need for targeted interventions that consider these intersecting identities to improve maternal health outcomes for both married and unmarried adolescent girls. For example, adolescent girls working in the garment sector face significant time constraints that limit their ability to access health services. One participant noted:

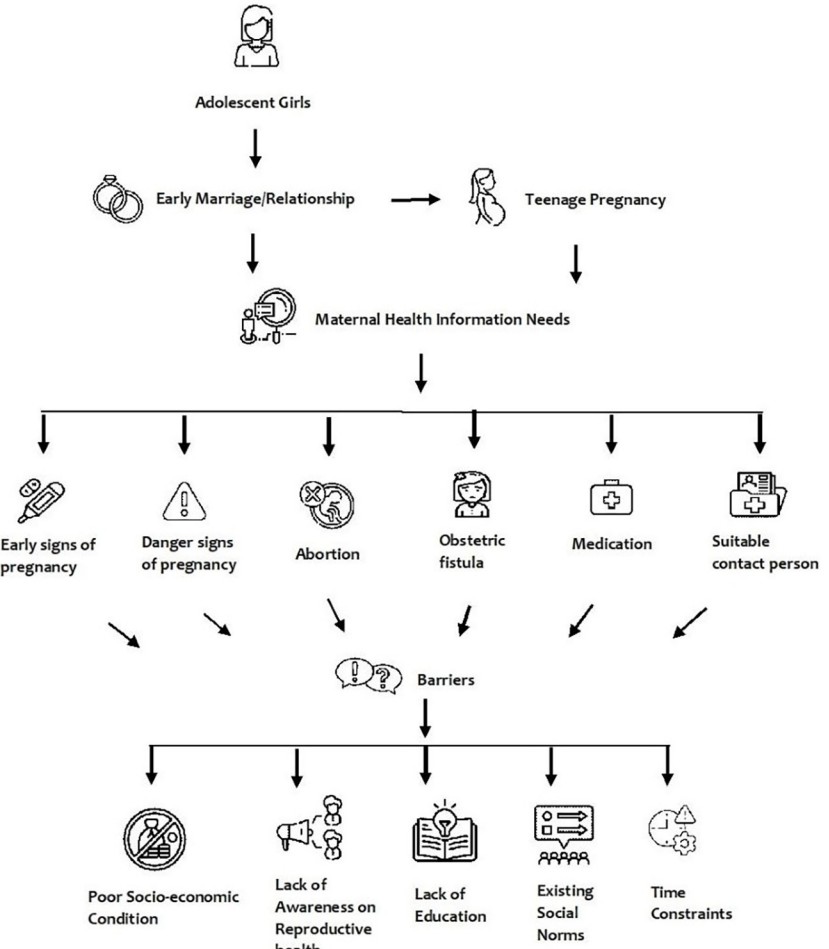

**Fig 1. Barriers to maternal health information seeking among adolescent girls.**

"*We can never get the government health facilities because of our office timing. The health service providers from the government do not come to us on our off days. And we do not stay at home on the other days. Most of us have to work overtime and return home late in the evening. So, how can we get the service, then?* (FGD 2, married adolescent girls)

As a result, they go to the local untrained health service providers like TBA and *Kabiraj* at a suitable time, as these practitioners are more available than the formal service providers. Moreover, informal health practitioners are less expensive compared to private health care services. Besides, the older family members (grandmother, mother, mother-in-law, aunt) suggest various ways to treat maternal health problems to the adolescents according to their life experiences.

The findings demonstrate that most adolescent girls, both married and unmarried, think that being a mother is an inevitable phase of any girl's life. However, they do not have the luxury to take this phase differently than any other typical diseases or health issues such as accidents or fractures, heart failure, strokes, etc., as most of them are garment workers and struggling with poverty. During the Focus Group Discussion, the married adolescent girls said,

"*These maternal care issues that you are asking, are for rich people. We, the poor, look forward to Allah (God) so that we can deliver the child safely at the minimum cost.*" (FGD 2, married adolescent girls)

Again, in the FGDs and IDIs, the unmarried adolescents mentioned some common barriers in their process of acquiring knowledge about maternal health. Most of them said that they did not think of inquiring about these issues before their participation in focus group discussions and interviews with us. As per their perception, they would avowedly learn about pregnancy, maternal health, etc. after marriage. That is the reason they do not bother much about these. One IDI participant stated,

"*How am I supposed to ask someone about these issues? These are very private matters. And I am not married yet. What would they think of me?*" (IDI 4, unmarried adolescent girl, age 18)

Majority of the FGD participants, too, illustrated that seeking maternal health information is not a task for unmarried adolescents. They said,

"*One, who will go to or show interest in maternal health care, will be treated as 'Charitrahin' (Characterless) in our society. These are needed after the marriage, and in our society, the older female members help the newly married girls with this information and share their experiences.*" (FGD 2, married adolescent girls)

On the other hand, the barriers faced by married adolescents were a bit different. Some of them argued that they only started to know from their in-laws about sexual and reproductive health issues, but not enough to bear a child and give birth. They strongly claimed that they wanted to understand the health conditions regarding pregnancy in a more detailed manner. But they hardly had any time or opportunity to do that, as she has no such agency for making decisions. A married girl shared her grief regarding her pregnancy during the in-depth interview. She said,

"*I did not want to have a baby so early. I wanted to take more time. Even before settling properly into this new family, I got pregnant. I talked to my husband about this and convinced him. But my mother-in-law did not agree. She insisted that my husband needed to convince me to have a baby at any cost. According to her, without having a child immediately after marriage, a girl cannot lead a conjugal life properly.*" (IDI 12, married adolescent girl, age 18)

Consequently, they seldom feel the necessity to seek information about maternal health. The socio-cultural context within which they live do not pay much attention to adolescents and the necessity of their access to proper knowledge about reproductive health. When we asked if they had attended any awareness-building programs on maternal or reproductive health, we got negative responses. Both the unmarried and married adolescents agreed that they hardly get to know about any awareness-building programs or campaigns. In the slum, the findings show that adolescents from Sattola slum only received one awareness program on adolescent health from a social development organization. That, too, was arranged two years ago, and because there was no refresher training arrangement, the adolescents could hardly remember the key messages from that program.

Adolescent girls' information-seeking behavior has been found to be influenced by their low levels of education. Participants stated that due to their low levels of education, they

**Table 6. Component and response of social norms.**

| Sl. | Components of Norm | Definition | Response from MAGs | Response from UAGs |
|---|---|---|---|---|
| 1. | Empirical Expectation | What I think others do | 1. MAGs from richer families get the information from formal sources | 1. UAG do not seek maternal healthcare information |
| 2. | Normative Expectation | What I think others expect me to do (what I should do according to other) | 1. Elder female members are expected to ask for maternal health care information by MAGs<br><br>2. Girls should have baby immediately after marriage | 1. UAGs do not need to know about maternal health care information |
| 3. | Sanction | Expected reaction of other (to the behavior)–specifically others whose opinions matter to me. | 1. People would say 'nashta' (Person with bad characteristics) if not carry babies earlier | 1. People would term 'besshya' (prostitute) if anyone come to know that an UAG seek maternal healthcare information |
| 4. | Sensitivity to Sanctions | Do sanctions matter for behavior? If there is a negative reaction from others (negative sanction), would the main character change their behavior in the future? | 1. Rich MAGs can visit formal information source<br><br>2. Poor MAGs cannot visit formal information source | 1. Neither rich nor poor UAGs can visit formal information source to seek these information |
| 5. | Exceptions | Under what conditions, it is okay to break the norm? | 1. If have rich family background<br><br>2. If have educated family background | 1. No exception is found |

sometimes have difficulty understanding written information sources. For example, it has been reported that some women cannot read and comprehend maternal health facts presented in print format due to illiteracy.

**Social norms.** Using vignettes during conducting IDIs and FGDs, we found that social norms are not the same, always, for MAGs and UAGs due to their marital condition. However, some similar norms are also found in some particular cases (Table 6).

The typical practice, known as empirical expectation, according to SNAP framework, in case of MAGs are MAGs from rich background can avail maternal health care information from formal sources like hospital and visiting doctors' chamber. In contrast, the UAGs do not seek information regardless their family's economic condition. Peoples expect that MAG should ask to their elder female members, such as mother, mother in laws, elder sister, and sister in laws, if they require any maternal health information. The core reason behind this is economic insolvency and privacy. Another reason is- if MAG ask extruders, the in-laws and neighbors would think that she does not want to have baby immediately after marriage where the expected behavior in community is any MAG should conceive in the very first year of marriage. If not, she is engaged with

> "*This is a private matter. We are not expected to ask about maternal health information anyone who is not our kin, as there are many family decisions embedded with this, such as keeping or not keeping the baby. Moreover, if a baby is aborted for any reasons, people will talk negative about the MAG or her family.*" (IDI 8, married adolescent girl, age 17)

On the other hand, UAGs cannot inquire these information as community people might think that the girl might get pregnant. Otherwise, why any UAG would would ask that.

> "*People would say that the UAG is 'besshya' (prostitute) if anyone come to know that an UAG is seeking maternal healthcare information.* (IDI 5, unmarried adolescent girl, age 17)

> "*People will count that UAG as pregnant, and spread rumor about her. It would be harmful for the girl and the family to arrange marriage for that girl*" (FGD 2, married adolescent girls)

Negative reactions or sanctions gives terrific feeling to MAGs and UAGs and their family. When any MAG listen that people from her community calls her "Nashta" (Person with bad characteristics) if not carries baby earlier, she feels mental pressure because of this "stigmatization" and often decide to change her decisions regardless their rich or poor background. Again, in case of the UAGs, findings revealed that people would call them 'besshya' (prostitute) if anyone came to know that an UAG seek maternal healthcare information.

*"It feels very bad when people use terms like 'besshya' (prostitute) without knowing the actual reason of seeking maternal health care."* (IDI 6, unmarried adolescent girl, age 17)

We found the existing social norms in the studied slum quite strong as economy is one of the crucial factors to have maternal health care information from any formal source. The findings demonstrate that only MAGs from rich family backgrounds can make some deviance of the norms of asking elder members about maternal health care information and carrying baby in the first year of marriage. They can visit formal sources and carry babies late. However, the UAGs cannot set example of exceptions whether from poor or rich background which indicates that social norms are more dominant to the UAGs in comparison to MAGs in the studied slum.

## Discussion

The study intended to explore the health information-seeking behavior of adolescent girls of reproductive age in the slum context of Bangladesh. It is clear from the data that the necessities for reliable maternal health information are not being met in the studied slums, which are consequently hampered by constraints like social norms and stigma, a lack of education and awareness, the timing of service provision, and so forth (Fig 1). The findings of this study highlight several critical aspects of maternal health information needs among adolescent girls in urban slums, revealing new insights and differences from previous research, particularly regarding gender dynamics and intersectional identities.

The unfamiliarity of participants, especially first-time adolescent mothers, with early pregnancy symptoms and associated risks underscores the low awareness about maternal health in the studied slum areas. This is consistent with wider studies in sub-Saharan Africa and South Asia and globally where a multi-conceptual background to pregnancy health and danger signs is associated with increased maternal morbidity and mortality [25]. Like with the adolescents in our study, they sourced maternal health information mainly from unverified sources, which has also been found in rural India and Nigeria due to lack of access to credible health information [26–28]. These parallels suggest that low awareness and reliance on unverified sources are common challenges in under-resourced settings.

Our study results are in contrast with other research works especially of Kamal et al (2015) and Haider et al (2023) [29, 30]. Evidence was also reported from similar settings, including Schuler et al. (2006) in Bangladesh, where logistical barriers such as time and service inadequacy were experienced were the leading causes of inquiries about non-certified knowledge [31]. On the contrary, our findings suggest that it is the time poverty and failure of mobilization within that proposed to be driving youth to questionable sources. This disparity demonstrates the need of context-appropriate interventions to remove barriers to access to information regarding maternal health.

The research findings of this study confirm the study by Zia et al. (2021) emphasizes that stigma results in significant underreporting of unsafe and clandestine abortions, especially in sub-Saharan Africa [32]. Such shame and stigma together result in an important psychosocial

burden on the lives of young women experiencing abortion and can justify the moral and frank ideological dilemmas encountered by them. This study outlines specific slum characteristics related to social dynamics. Findings say that UAGs in the slums are hesitant to disclose abortion because there are further violent social repercussions. Identification through a successful abortion would expose a UAG to a lifetime of societal censure and escalation by rendering her unmarriageable and putting her and her family at risk for public ostracism and violence. It corresponds with the wider impression of stigma-related difficulties highlighted in previous literature as well [33]. Additionally, our study suggests that interventions to reduce abortion stigma must be tailored to the complex social norms and consequences of abortion stigma for women living in predominantly slums, which highlights a unique area for future research and intervention development [32, 33].

The research also revealed a profound unawareness of family planning and obstetric fistula among adolescent girls. These findings are consistent with a body of evidence from Ethiopia and Kenya which shows how entrenched cultural beliefs and poor sexual and reproductive health education perpetuate misconceptions about reproductive health [34, 35]. Although we already know that previous research has shown misinformation about such issues [36], our data shows how such knowledge gaps are exacerbated by social stigma and certain cultural beliefs of witchcraft as a cause of obstetric fistula. Additionally, this ignorance getting in the way of teenagers being able to use the health service when they are there to be used, as well as continuing the problem practices and beliefs. The results seem to call for the development of more specific educational programs on family planning, which should aim at responding to women's practical and strategic gender needs by disapproving unhelpful myths and providing reinforcing messages [37].

Health information seeking from informal sources such as elder female relatives is not only common in the studied slums but also in patriarchal societies around the world. The studies in Afghanistan and Nepal show that, accompanied by ego constructs and social norms, women have less mobility and less usage of formal health services than men, and women often rely on family members for their health information [38, 39]. Taken together, these findings suggest that community health workers and informal networks should be integrated into more formally directed health education programs to ensure valid information is spread.

## Strengths and limitations

The strength of this study lies in its use of open-ended qualitative research which allowed the emergence of key issues in maternal health. The use of SNAP framework as an interpretive tool provided insights into the communities' realities that have been diminished through the lens of dominant but not localized theories [14]. On the other hand, theory of social norms helped us to underlying factors of adolescents' unwillingness or barriers to seek maternal healthcare information. Using these two approaches, the study draws on diverse perspectives and experiences of married and unmarried adolescent girls as it allowed for an exploration of latent themes, for instance, husband's and senior female members of in-laws' influence over their healthcare even with gender expectations that hold sway in communities.

While this study offers important perspectives on maternal health care, the reported findings should be interpreted in light of a number of limitations. The degree to which probing was employed as a technique by the research assistants may have influenced the responses generated by participants. However, it is noteworthy that the academic background and training of the field researchers, Anthropology, was helpful in mitigating the 'otherness' during the data collection [40]. Another important limitation is that by design, purposively selected participants' perceptions might not represent the realities of every adolescent in other slums of

Dhaka or any other urban slum of the country. Lastly, the study did not engage female senior members as it focused on the adolescents' perception. Including this category might illustrate more interesting findings.

## Policy recommendations and future research

Findings suggest three core recommendations. First, to organize and arrange the health service for adolescents by considering their occupational engagement. More specifically, before providing health service, it is essential to ensure the proper information is disseminated properly. If not, the overall health problem solving goal would be facing difficulties. Second, only targeting the adolescents for awareness session is not enough. As the husbands and in-law members are the key decision makers, it must be ensured that these people are well-aware of the risks of lack of maternal health care information. Third, programs targeting to reduce maternal mortality rate, adolescent sexual and reproductive health and rights (SRHR) should focus more on social norms and stigma as these are embedded in-depth into our society and culture.

For further research, in our opinion, more studies should be done on context-specific social norms. In Bangladesh context, there is a dearth of research on social norms regarding maternal health information seeking behavior and other SRHR issues using social norm's framework.

## Conclusion

The primary goals of this study were to examine the information needs of adolescent girls on maternal health, identify trustworthy sources, and determine how social norms impede their ability to get maternal health information. Study findings demonstrate the influence of established factors as well as new insights into the gendered intersectional dimensions of maternal health information-seeking behaviors among adolescent girls in urban slums. The findings and analysis illustrate those adolescent girls, both married and unmarried, hardly have formal knowledge regarding maternal health information. The existing knowledge is inadequate and received only from informal sources, which shapes behavior in a biased and vulnerable way. This appears more precarious in the cases of the slum girls, as they are married at an early age by their parents to ensure their social security. The existing barriers, like health service timing, need to be altered. As the adolescents are engaged with a variety of jobs and work late at night almost six days (only one off-day) in a week, health services and awareness programs should be reconsidered and redesigned. Moreover, some interventions may also be initiated to change the existing social norms and stigma regarding the practice of receiving health information from informal sources around this group, as this is one of the key drivers for creating constraints ensuring better health information-seeking behavior. Policymakers can utilize this study as it explains how a social norm lens can help to integrate practical and strategic gender needs and improve programmatic efforts to enhance and scale up health education programs for young adolescents with a specific focus on the intersectional identities of these adolescents. These programs need to be designed to correct false information targeting adolescent girls, engage men in maternal health education, and contest broader, patriarchal norms that restrict women's agency and mobility. On the other hand, academicians might be encouraged to engage themselves in social norm analysis to explore better explanations of any health problem in the context of Bangladesh where such research is scant.

## Supporting information

**S1 Table. Consolidated criteria for reporting qualitative studies (COREQ) checklist.**
(DOCX)

## Acknowledgments

We are thankful to all the participants in this study who gave their valuable time after their restless day-long work. Also, we would like to express our heartfelt gratitude to the reviewers for their valuable input.

## Author Contributions

**Conceptualization:** Md. Ashraful Haque.

**Data curation:** Rabeena Sultana Ananna, Nayeem Hasan.

**Formal analysis:** Md. Ashraful Haque, Rabeena Sultana Ananna, Mst. Farhana Aktar, A. F. M. Zakaria.

**Investigation:** Md. Ashraful Haque.

**Methodology:** Md. Ashraful Haque, Rabeena Sultana Ananna, Nayeem Hasan, A. F. M. Zakaria.

**Project administration:** Mst. Farhana Aktar.

**Software:** Md. Ashraful Haque.

**Supervision:** Md. Ashraful Haque, A. F. M. Zakaria.

**Validation:** Md. Ashraful Haque, Rabeena Sultana Ananna, Mst. Farhana Aktar.

**Visualization:** Md. Ashraful Haque, Mst. Farhana Aktar, A. F. M. Zakaria.

**Writing – original draft:** Md. Ashraful Haque, Rabeena Sultana Ananna, Nayeem Hasan.

**Writing – review & editing:** Md. Ashraful Haque, A. F. M. Zakaria.

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
