## [Decision Letter · Decision Letter 0]

1 May 2024

PONE-D-23-34519A Qualitative Study on the Maternal Health Information-Seeking Behavior among Adolescent Girls of Reproductive Age in a Slum of Dhaka, BangladeshPLOS ONE

Dear Dr. Haque,

Thank you for submitting your manuscript to PLOS ONE. After careful consideration, we feel that it has merit but does not fully meet PLOS ONE’s publication criteria as it currently stands. Therefore, we invite you to submit a revised version of the manuscript that addresses the points raised during the review process.

 The revision will be sent out to review again.  Please submit your revised manuscript by Jun 15 2024 11:59PM. If you will need more time than this to complete your revisions, please reply to this message or contact the journal office at plosone@plos.org. Please include the following items when submitting your revised manuscript:A rebuttal letter that responds to each point raised by the academic editor and reviewer(s). You should upload this letter as a separate file labeled 'Response to Reviewers'.A marked-up copy of your manuscript that highlights changes made to the original version. You should upload this as a separate file labeled 'Revised Manuscript with Track Changes'.An unmarked version of your revised paper without tracked changes. You should upload this as a separate file labeled 'Manuscript'.

We look forward to receiving your revised manuscript.

Kind regards,

Sanzida Akhter, PhD

Academic Editor

PLOS ONE

Reviewers' comments:

Reviewer's Responses to Questions

**Comments to the Author**

1. Is the manuscript technically sound, and do the data support the conclusions?

Reviewer #1: Yes

Reviewer #2: Yes

2. Has the statistical analysis been performed appropriately and rigorously? 

Reviewer #1: N/A

Reviewer #2: N/A

3. Have the authors made all data underlying the findings in their manuscript fully available?

Reviewer #1: Yes

Reviewer #2: Yes

4. Is the manuscript presented in an intelligible fashion and written in standard English?

Reviewer #1: Yes

Reviewer #2: No

5. Review Comments to the Author

Reviewer #1: Review of study entitled “A Qualitative Study on the Maternal Health Information-Seeking Behavior among Adolescent Girls of Reproductive Age in a Slum of Dhaka, Bangladesh”

Reviewers Comment

1. This is a crucial topic in addressing the unmet needs of adolescents Maternal Health Information-Seeking Behavior among Adolescent Girls of Reproductive Age in a Slum in Dhaka, Bangladesh but the findings of the study were not elaborate enough even though exploratory design was employed.

2. Language: Authors need to improve on the grammar of the manuscript. Sometimes authors use ‘previous studies’ but refer to only one study, see introduction, methodology sections etcetera. Also, authors use certain terms that are not familiar in the Englush language.

3. The title: “A Qualitative Study on the Maternal Health Information-Seeking Behavior among Adolescent Girls of Reproductive Age in a Slum of Dhaka, Bangladesh”. This is confusing, what does adolescent girls of reproductive age mean?

4. Abstract: Authors do not indicate which category of participants the IDIs and FGDs were conducted among. The results section lacks completeness and a logical flow as well as how and why the source of information is not considered a problem.

5. The conclusion in the abstract appears more like a recommendation. The authors can review the abstract conclusion.

6. Introduction: The introduction is not focused and lacks a logical flow. Authors need to be more focused.

7. The methods section is scanty. Consider using the Consolidated criteria for reporting qualitative research (COREQ) to report on the missing aspects.

a. Also, the design is mentioned twice

b. Authors failed to explain the type of purposive sampling technique that was used

c. Authors fail to report on how participants were recruited, the focus of the IDI, KII and FGD guides, the pharmacists are missing from the table

d. The authors fail to explain the reasons for choosing the study design and why pharmacists were included. Generally, there is poor or lack of justification for decisions made in the methodology section

8. The narration in the results section sometimes appears to be more of commentary than reporting qualitative results from study participants eg first paragraph on page 8.

a. I could not comprehend table 4

b. The authors should improve on the structure. Sometimes a topic is introduced without any warning or when it is not in relation to information in that section for instance the narration on fistula, should have a separate section, it does not fit well, where it is currently.

9. This is an important topic that extends beyond the study area, but the literature used in the discussion section is very scanty. The authors fail to make their paper relevant to the broader society by not engaging extensively in literature pertaining to other parts of the world.

10. The recommendation is not adequate and not targeted to specific policymakers and other stakeholders

11. The recommendation should be separated from the conclusion

12. The study did not talk about the available health facilities in Sattola and its surrounding

13. To obtain a more accurate picture of the situation in Sattola Slum, it would be great if health professionals were also included in the study participants to know their engagement efforts and barriers to reaching out to adolescent females as well as unmarried and married adolescent males.

Reviewer #2: Comments

Overall, the authors tried to fulfil the objectives of the study. However, the authors could not address gender issues properly when they were analysing things. At some point, the gender lens is missing from this article. They mentioned many things. However, this was not properly aligned with gender terms like gender needs (Practical and strategic gender needs), and social stigma. Also, they mentioned different kinds of identities in demographic profiles as well. However, the analysis part could not give the proper justice to these different kinds of intersectional identities. They should have provided a deeper analysis of how married women are not free to make any decision according to their will, and why unmarried adolescent girls are not willing to talk. More details and a thorough analysis of this part with a combination of primary and secondary data would be expected.

They interviewed a few numbers of IDI among unmarried girls. Therefore, it’s not okay to perceive that unmarried participants were not willing to share their abortion history. Probably, they did not have. As they mentioned, “Though KII participants clarified that they know of some cases of abortion by unmarried adolescent girls, no unmarried IDI participant was keen to share those pieces of information with us.” The authors need to explain this statement more. Why do they think like that? What kinds of stigma do they face from the society? Also, they need to back up their primary data providing secondary data as well. In the analysis section, primary data is not getting enough backup or supported by secondary data.

Without having the proper knowledge of sexual intercourse, they can become pregnant. At the same time, they can get STDs as well. However, this was not mentioned in this article. The authors could come up with a recommendation like providing door-to-door health services on weekends as well. The authors could have mentioned the discussion from the “ Three Delay Model” and aligned it with their study. At some point, I felt it was necessary to talk with their family members like husband/partner to understand their conjugal understanding and decision-making process. It was also necessary to talk with parents or in-laws to focus on the stigmatized views of society.

In the discussion part, the authors tried to mention how this study is similar to other studies but it was also necessary to mention how this study is different from other studies, and what new pieces of knowledge or information are produced by this study. Is it bringing something new for us? Also, analysing the position and condition of these adolescent girls in a patriarchal setup like a slum area was not investigated properly. As this study is concerned with marginal women, it is necessary to have a gender lens to analyse the findings properly. How women were controlled, how their mobility was controlled, how they were bound to act according to imposed in-laws' decisions, and how these facts are shaping adolescents' maternal health information-seeking behaviour. These things should be discussed in the author's write-up. Overall, the theoretical and conceptual part was ignored in this study.

Furthermore, the authors should be more careful about sentence formation. “The results of our study are different from those of earlier studies (18) which found that obstacles like long travel times to medical facilities and substandard services on maternal health from outreach programs force women to approach unauthorized providers to accumulate information in the slums, which are situated in a city.” It is suggested to avoid long lines like this as this is a little bit confusing for readers. Also, the conclusion should bring more information or recommendations from this study.

6. PLOS authors have the option to publish the peer review history of their article (what does this mean?). If published, this will include your full peer review and any attached files.

Reviewer #1: No

Reviewer #2: **Yes: **Kuntala Chowdhury

---

## [Author Response · Author response to Decision Letter 0]

14 Jun 2024

June 14, 2024

Sanzida Akhter, PhD

Academic Editor

PLOS ONE

Manuscript Number: PONE-D-23-34519

Title: A Qualitative Study on the Maternal Health Information-Seeking Behavior among Adolescent Girls of Reproductive Age in a Slum of Dhaka, Bangladesh

Dear Editor,

Thank you for allowing us to resubmit the revised manuscript. We appreciate the time and effort that you and the reviewers have dedicated to providing us with your valuable feedback on our manuscript. We are truly grateful to the reviewers for their insightful comments on our paper. It was a great learning for us. We believe these suggestions will help improve the quality of the manuscript. Therefore, we have incorporated changes to reflect all the suggestions provided by the reviewers. We have used track changes option within the manuscript to highlight the changes.

In addition, this is to inform you that as some of our affiliation has changed by this time, we have changed that in the portal and the title page. 

Here is the point by point responses to the reviewers’ comments and concerns: 

Response to Reviewer #1: 

Reviewer #1: Review of study entitled “A Qualitative Study on the Maternal Health Information-Seeking Behavior among Adolescent Girls of Reproductive Age in a Slum of Dhaka, Bangladesh”

Reviewers Comment

Reviewer point #1: This is a crucial topic in addressing the unmet needs of adolescents Maternal Health Information-Seeking Behavior among Adolescent Girls of Reproductive Age in a Slum in Dhaka, Bangladesh but the findings of the study were not elaborate enough even though exploratory design was employed. 

Author response #1: Thanks for your in-depth feedback. We have tried to incorporate elaboration for your reference. As this paper was a maiden paper from the team, we faced some struggles earlier in coordinating and collaborating. 

Major changes in this version:

1. Theoretical model is added

2. Methods are re-written

3. Grammar is revised

4. Results are re-written in elaborative manner

5. Structure is revised

Reviewer point #2. Language: Authors need to improve on the grammar of the manuscript. Sometimes authors use ‘previous studies’ but refer to only one study, see introduction, methodology sections etcetera. Also, authors use certain terms that are not familiar in the Englush language.

Author response #2: We agree on this comment. The whole paper has been reviewed and find grammatical and logical flaws. The concern of “Sometimes authors use ‘previous studies’ but refer to only one study” were addressed. 

Reviewer point #3. The title: “A Qualitative Study on the Maternal Health Information-Seeking Behavior among Adolescent Girls of Reproductive Age in a Slum of Dhaka, Bangladesh”. This is confusing, what does adolescent girls of reproductive age mean?

Author response #3: We put the words ‘adolescent girls of reproductive age’ by considering the mostly known definition of adolescents, who are from 10-19 years, and the reproductive age of a women, which is 15-49 years. In such sense, adolescent girls of reproductive age refer to the girl who are between 15 and 19 years old. We set this explanation in fourth sentence of Study Design in Methodology section.

However, we have changed our research title from A Qualitative Study on the Maternal Health Information-Seeking Behavior among Adolescent Girls of Reproductive Age in a Slum of Dhaka, Bangladesh to Social norms and maternal health information-seeking behavior among adolescent girls of reproductive age: A qualitative study in a slum of Bangladesh.

Reviewer point #4. Abstract: Authors do not indicate which category of participants the IDIs and FGDs were conducted among. The results section lacks completeness and a logical flow as well as how and why the source of information is not considered a problem.

Author response #4: Categories are mentioned in Abstract’s Method section. Also, we have included theoretical framework as the foundation for the study's conduct and design. 

Edited version:4

Methods: Adopting an explorative qualitative research approach, we collected data from purposively selected married and unmarried adolescent girls (15-19 years old) of different occupation by implying 12 in-depth interviews (IDIs), two Focus Group Discussions (FGDs) with same categories employed for IDIs, and two Key Informant Interviews (KIIs) with a traditional birth attendant and a local pharmacist. Furthermore, the data were subjected to thematic analysis. Care’s Social Norms Analysis Plot (SNAP) framework were undertaken as interpretative tool for data that was emerging rather than serving as the foundation for the study's conduct and design. Thematic analysis were followed to analyze primary data.

The Result section of Abstract is re-written by addressing reviewer’s comment. 

Results: Majority reported that adolescent girls need professional healthcare providers in their area who would work according to their work schedule as most of the girls are engaged in income generating work for about 9-11 hours, and the scope of work (daily wagers) hardly supports ‘leave with pay’. In the present context, the most of the adolescent girls rely on elder female family members, traditional birth attendants, and local pharmacists for maternal health information. These unverified sources prescribe treatment such as medicine, food, and herbal products without formal expertise which causes life risks for adolescent mothers. Additionally, existing social norms and stigma have a bigger impact on the maternal health information seeking of adolescent girls.

Reviewer point #5. The conclusion in the abstract appears more like a recommendation. The authors can review the abstract conclusion.

Author response #5: The Conclusion section is re-written by addressing reviewer’s comment. 

Conclusion: The study offers a different narrative in the discussion of maternal health information seeking behavior, specifically how social norms and stigma are preventing slum adolescents to get their desired health information. The results might be useful for informing policy and program designing to ensure better health outcome for marginalized adolescents.

Reviewer point #6. Introduction: The introduction is not focused and lacks a logical flow. Authors need to be more focused.

Author response #6: We have tried to rewrite the Introduction section, accordingly.

INTRODUCTION

Maternal health is a worldwide issue, even though it was reduced by 38% from 451,000 in the year 2000 to 295,000 deaths in 2017. This is a particular problem for developing countries where 94% of maternal deaths occur (1). In Bangladesh, reducing maternal mortality is a national priority, supported by policies like the Bangladesh National Health Policy 2011 (2); as long as with the Bangladesh National Strategy for Maternal Health 2019-2030 (3). The issues of maternal health are in the attempts to reduce all the factors that negatively affect the physical condition and social position of women and mothers. At the present moment, the number of the mortality rates had reduced in comparison to 2000 when it was 434 per 100,000 live births, and in 2017, it was 173, whereas the current goal for 2030 is to make it lower than 70 (1).

Bangladesh has one of the highest adolescent fertility rates in the Asia Pacific, with 128 births per 1000 girls aged 15 to 19 (4). Adolescent girls are married off 3-4 years earlier than the legal marriage age of 18. Married adolescent girls go through unavoidable social and family pressure to get pregnant soon after marriage as proof of their fertility (5). Their incapacity to obtain family planning (FP) and reproductive health (RH) services encourages early childbirth and marriage (6). In contrast to rural women, women living in urban regions are more likely to have made four or more antenatal visits (59% vs. 43%) (7). Again, the health-seeking behavior of people living in cities varies. Evidence illustrates that 20% of teenage mothers gave birth unintendedly (5), and unwanted pregnancies are more than twice as common among married adolescent girls in Bangladesh's slums as in non-slum areas (6). Previous study demonstrates that Bangladeshi slum had a maternal mortality rate of 294 deaths per 1000,000 live births and a neonatal mortality rate of 43 deaths per 1,000 live births (8).

Access to timely and relevant maternal health information is crucial for informed decision-making and reducing maternal morbidity and mortality. Accurate maternal health information promotes positive health behaviors and better maternal health outcomes (9). Delays in seeking appropriate care during pregnancy significantly increase maternal risks. While some studies have addressed maternal and neonatal mortality in slums, there is a notable gap in research on the health information-seeking behavior of adolescent girls in Bangladesh's urban slums. This is critical, given that slums host over 5.7 million people, approximately 3.8% of the national population, most of whom are migrants seeking better economic opportunities (10). Addressing the maternal health information needs of this marginalized group is essential for improving maternal health outcomes in Bangladesh.

 Care’s Social Norms Analysis Plot (SNAP) Framework was developed to measure the nature of specific social norms and their influence and offers a useful framework to examine the initial reactions to a social norms activity. Empirical expectations, normative expectations, sanctions, sensitivity to sanctions, and exceptions are five key constructs to understand one’s act or decision-making (11). Although studies exist on overall unexpected pregnancy and contraceptive use among married adolescent girls in Bangladesh, data on particularly vulnerable groups (as our study examines both married and unmarried adolescents from an urban slum) of adolescents is scant. In addition, to our best knowledge, there is no research done yet to understand the maternal health care information seeking in social norm perspective. Therefore, we aim to explore the maternal health information-seeking behavior of adolescent girls of reproductive age in a slum through the lens of social norms. The primary goals of this study were to identify adolescent girls' maternal health information needs during their reproductive years and to learn about how social norms working as the obstacles they face when seeking information related to maternal health.

Reviewer point #7. The methods section is scanty. Consider using the Consolidated criteria for reporting qualitative research (COREQ) to report on the missing aspects.

Author response #7: Thanks. We have incorporated Consolidated criteria for reporting qualitative research (COREQ) to report on the missing aspects. The overall method section is re-written and currently we have Study Design, Study Setting, Sampling and Recruitment, and Data Collection Procedure, Data Analysis, Ethical Consideration and Ethical Clearance sub-section.

Reviewer point #7 (a). Also, the design is mentioned twice

Author response #7 (a): Thanks. We have addressed the concern.

Reviewer point #7 (b). Authors failed to explain the type of purposive sampling technique that was used

Author response #7 (b): The explanation is given.

Sampling and Recruitment

The study used purposive sampling method to recruit study participants as per convenience due to the variety of occupational engagement of the study population. To ensure confidentiality when discussing delicate subjects, participants were assigned with same-sex data collectors. The data collectors were trained anthropologists and geographer. The field researchers received three days of training from the lead investigators, with a focus on sensitive research involving human subjects and notably adolescent subjects conducted in an ethical manner. To maximize comprehension and enable correct response documentation, the IDI, FGD and KII guides with vignettes were tested, revised, and collaboratively translated into Bengali and back into English. 

Reviewer point #7 (c). Authors fail to report on how participants were recruited, the focus of the IDI, KII and FGD guides, the pharmacists are missing from the table

Author response #7 (c): We have added the ways we followed to recruit participants. Please, see the Methodology section

Reviewer point #7 (d). The authors fail to explain the reasons for choosing the study design and why pharmacists were included. Generally, there is poor or lack of justification for decisions made in the methodology section

Author response #7 (d): In the edited version of the paper, we have explained the justification. Also, we added why we recruit the KIIs. 

Study Design

We undertook an exploratory qualitative method, a widely used methodological approach, especially within practice disciplines (12), to have a deeper understanding of the maternal health information-seeking behavior among adolescent girls of reproductive age in a slum of Dhaka, Bangladesh. We used the terms- adolescent girls of reproductive age, by considering the mostly known definition of adolescents, who are from 10-19 years, and the reproductive age of a women, which is 15-49 years, according to WHO (13, 14). In such sense, adolescent girls of reproductive age refer to the girl who are between 15 and 19 years old. This approach was selected for this study due to its nature of nature of investigating research objectives by exploratory questions (15). The lack of in-depth research on maternal healthcare access among this age group, living in slums, led us to a need for an in-depth exploratory study. For this, we selected in-depth interview (IDI), focus group discussion (FGD) and key informant interview (KII) (Table 1), to provide a data source for triangulation. IDI participants provided subjective viewpoints and FGD participants helped us to understand the community perspective. Participants of KII facilitated to get objective viewpoints of the data collected by other methods. We conducted IDIs with six married adolescent girls (MAG) and six unmarried adolescent girls (UAG) and two FGDs with MFAG and UFAG. Among two KII participants, one was local traditional birth attendant (TBA) and the other was a pharmacist by profession. TBA was recruited as majority of the study participants feel comfortable to visit her. On the other hand, pharmacist is the key person to the slum to take medicine. Participants consider them as doctor and take medicine from them without prescription, rather just telling health problems. Despite UAGs’ and MAGs’ access to Kabiraj (faith healer), we could not interview him as he was outside of the slum due to personal reasons. Furthermore, we approached formal health service providers whose facility was closer to the slum, however, as he reported that he received any adolescent girl from the studied slum as patient, we did not consider them as KII.

Reviewer point #8. The narration in the results section sometimes appears to be more of commentary than reporting qualitative results from study participants eg first paragraph on page 8.

Author response #8: Thanks. We have rewritten the paragraph.

Reviewer point #8 (a). I could not comprehend table 4

Author response #8 (a): We agree with you. The table is removed. We tried to illustrate the consequences of unintended pregnancy for the lack of knowledge, earlier.

Reviewer point #8 (b). The authors should improve on the structure. Sometimes a topic is introduced without any warning or when it is not in relation to information in that section for instance the narration on fistula, should have a separate section, it does not fit well, where it is currently.

Author response #8 (b) We focused on structure in the revised version.

Reviewer point #9. This is an important topic that extends beyond the study area, but the literature used in the discussion section is very scanty. The authors fail to make their paper relevant to the broader society by not engaging extensively in literature pertaining to other parts of the world.

Author response #9: We have re-written the discussion by relating other scholarly research work all over the word

Reviewer point #10. The recommendation is not adequate and not targeted to specific policymakers and other stakeholders

Author response #1

---

## [Decision Letter · Decision Letter 1]

2 Oct 2024

PONE-D-23-34519R1Social norms and maternal health information-seeking behavior among adolescent girls of reproductive age: A qualitative study in a slum of BangladeshPLOS ONE

Dear Dr. Haque,

Thank you for submitting your manuscript to PLOS ONE. After careful consideration, we feel that it has merit but does not fully meet PLOS ONE’s publication criteria as it currently stands. Therefore, we invite you to submit a revised version of the manuscript that addresses the points raised during the review process.

**ACADEMIC EDITOR: **

**Kindly incorporate the reviews suggested by the reviewers.**

We look forward to receiving your revised manuscript.

Kind regards,

Ranjit Kumar Dehury

Academic Editor

PLOS ONE

**Journal Requirements:**

Reviewers' comments:

Reviewer's Responses to Questions

**Comments to the Author**

1. If the authors have adequately addressed your comments raised in a previous round of review and you feel that this manuscript is now acceptable for publication, you may indicate that here to bypass the “Comments to the Author” section, enter your conflict of interest statement in the “Confidential to Editor” section, and submit your "Accept" recommendation.

Reviewer #2: All comments have been addressed

Reviewer #3: (No Response)

2. Is the manuscript technically sound, and do the data support the conclusions?

Reviewer #2: Yes

Reviewer #3: Yes

3. Has the statistical analysis been performed appropriately and rigorously? 

Reviewer #2: N/A

Reviewer #3: N/A

4. Have the authors made all data underlying the findings in their manuscript fully available?

Reviewer #2: Yes

Reviewer #3: Yes

5. Is the manuscript presented in an intelligible fashion and written in standard English?

Reviewer #2: Yes

Reviewer #3: Yes

6. Review Comments to the Author

**Reviewer #2: **The authors addressed all my comments. In the rebuttal table, they also explained how they addressed my comments. I am satisfied with their current version of the work. From my side, this version of work is quite fine.

**Reviewer #3:** Manuscript ID: PONE-D-23-34519R1

Manuscript Title: Social norms and maternal health information-seeking behavior among adolescent girls of reproductive age: A qualitative study in a slum of Bangladesh

Dear Md. Ashraful Haque,

Thank you for the opportunity to review this manuscript. This is an interesting study that offers some insights into the maternal health of adolescent girls in an urban setting in Bangladesh. The issue of social norms and maternal health-seeking behavior among adolescent girls is an increasingly important subject, especially in light of the current global health complexities. The links to maternal health information seeking and influences of social norms among adolescent girls everyday ways of activities are also important and could contribute to the growing literature around this subject, especially in the urban slum context who are mostly migrated from the rural areas. However, I agree that the qualitative methods and sample size appear appropriate except for the KIIs number. The manuscript may be publishable, but I have a number of considerable concerns that should be addressed.

1. The title could be revised. ‘Adolescent girls of reproductive age’ is likely unnecessary, as adolescence represents the onset of reproductive ability, which might make the title concise. So, ‘reproductive age’ can be deleted. Additionally, the author line may be revised as 3 merging the numbers 3,4 are the same organization, ‘CARE Bangladesh’.

2. In abstract:

- The author mentioned ’12 IDIs’, ‘Two FGDs’, and ‘Two KIIs’. 12 may be write into words. Also write the age range into a bracket that may be written as ‘adolescent girls aged 15-19’, avoiding the bracket.

- KIIs with one traditional birth attendant and one local pharmacist. As the study was conducted in an urban slum, there may be chances to find a trained birth attendant along with a traditional one. Because, NGOs are implementing many maternal health projects for the slum dwellers in parallel with governments. Inclusion of trained birth attendants by interviewing may add more or new reflections and dimensions of data and strengthen the results.

- Further, ‘local pharmacist’ is not an appropriate term in the context of Bangladesh, as they are already identified as ‘drug sellers’ considering their academic background and training. See (Ahmed SM, Naher N, Hossain T, Rawal, LB. Exploring the status of retail private drug shops in Bangladesh and action points for developing an accredited drug shop model: a facility based cross-sectional study. J Pharm Policy Pract. 2017).

-Results section, started with the recommendations like ‘adolescents need professional healthcare’. It’s better to start with the maternal health information sources and a trusted one. Therefore, the existing social norms need to describe how they ‘affect’ maternal health of adolescent girls and why these social norms are being influences. Delete the word ‘impact’ as your study observed not long term results.

- In conclusion, the author should be aware of using judgmental decisions like ‘without……information, no girl or woman’ would be able to make crucial decisions…. Health. Please revise.

3. Introduction section, the first sentence statistics were outdated. You can find the updated WHO Maternal Mortality: Key Facts data using the link https://www.who.int/news-room/fact-sheets/detail/maternal-mortality. In the line 6 of this section, the author mentioned that ‘Bangladesh has not been able to eliminate maternal mortality’, is it possible? Better to write ‘reduce’ the number following the WHO standard or like wealthier countries. However, the author summarizes the literature on maternal mortality and maternal health in global and Bangladesh contexts, health consequence and increasing risks among the adolescent. In doing so, it would be useful to further discuss the healthcare practices/facilities, especially for the adolescent group with maternal health information and behavior of Bangladesh, the role of healthcare providers, family members before the aim of the study in the introduction section. As your results already explored and you include this in your discussion part of the elder family member involvement issues in relying for health information. In the fourth para, fifth line from last, the author describe ‘women in making educated decisions…’ , could you explain it. The next line, ‘we found no studies on health information-seeking….is also judgmental. Suggesting write ‘paucity’ or ‘limited’ studies or others. In the same line the author contextualized the issue well describing slum population which citation is outdated (Khatun F. et al. 2012). The author can review the World Bank data, and even can cite Hasan MZ, Hasan AMR, Rabbani AG, Selim MA, Mahmood SS. 2022. Knowledge, attitude, and practice of Bangladeshi urban slum dwellers…… PLOS Global Public Health that identified “Around 38% (62.5 million) of the total population of this country live in urban areas and about half of them live in slums.” Please try to cite upto 2020 except classic references.

4. Methods could be better justified. This topic is likely to have significant social acceptability bias, which needs to be acknowledged. Specifically-

- The author need to contextualize the urban slum and then ‘Sattola’ describing the situation of slum, and its population in more details in the study site section.

- I believe this may be a typing error as it took ‘one month’ in 2.1 section and then ‘two months’ in the section 2.2.

- The sample size was not justified scientifically. The author could follow the data saturation principles for data collection is required here to fit in more within the qualitative paradigm. Better to include one reference here on data saturation related past/current article.

- Since the author recruited 12 married and unmarried adolescent girls, TBA and pharmacist (drug seller), it is worth mentioning the participants’ inclusion and exclusion criteria. Moreover, authors could discuss the challenges including attitudinal dimension that the team faced during data collection. Recruiting trained birth attendant and elder family member is already suggested in abstract section.

- Data collection procedures were not described well. Author could describe the research team academic background, training and experiences, interview guidelines (semi-structured or others), guidelines developing procedures, field test, interview settings (slums dwellers shared room, present about the ‘place' conducted interviews), recordings, participation type (volunteer or with pay)…. I suggest to follow or maximize the COREQ 32 items and may cited (https://www.ncbi.nlm.nih.gov/books/NBK554122/) this or any other related article.

- This is unclear what did the ‘deductive code list’ entail? As the study design is qualitative, why authors followed the deductive way is conflicting. Please revise this or explain.

- Thematic analysis procedures need to be explain more and need cited. Second paragraph of data analysis, author mentioned ‘themes were developed and recording was done…. understanding’ is unclear. Can revise following a reference. This could be more reflection on this and on the strengths (arguments, justifications for why the author use) which method brings to the analysis. In addition, ‘to confirm the accuracy…. and ‘cross-matched’ is overlap with triangulation, can revise.

- In ethical issues, the author mentioned followed …. ‘high priorities’.. need to explain how. ‘All of the participants gave their written consent’ need to revise. As the study participants were the garment workers and sat in for interview on their day off, please emphasize and highlight on the interview time, compensation, privacy and confidentiality (as they live in tiny and shared rooms) of the study participants and their shared experiences that ensured in the entire study. Because, author already described the hectic office hours of the participants and day off may their priority to get adequate rest. ‘One ethical specialist’ was recruited in the team, what was his/her roles. The author may revise the sentence … ‘destroyed data’ as the manuscript is not published yet and might need go back thorough data again.

5. The results are interesting but not line with study aim completely. However, it requires some reorganization of the results are following-

3.1 In theme 1, the author mentioned ‘maternal health information needs for adolescent girls’ is kind of assessment but aim of study was to ‘understand the maternal health information-seeking behavior and then primary goal was assess….., the author can revise the result section following the aim of study. Like, health information sources, trusted sources, social norms that influences the behavior, influences of health information seeking behavior, barriers/challenges in seeking behavior, and then needs of health information.

3.3 Moreover, I would ask to include interview number in the quotes.

6. Discussion: Suggesting knowledge is ‘accurate’ isn't very helpful with a qualitative study with such a small sample (and with no additional data, e.g. observations etc), but instead you can reflect on how knowledge is articulated or, as you do in the paper, discussing the role of the family in supporting to provide right information. Following the current literatures in introduction and reorganize the results section, the discussion need to be improved.

7. Conclusion: The first lines in the conclusion do not give any information that was intended to be explored in the study. In conclusion, readers want to see key findings interlinked with the social norms and health information seeking behavior that are based on empirical evidence, and then link those to policy recommendations.

7. PLOS authors have the option to publish the peer review history of their article (what does this mean?). If published, this will include your full peer review and any attached files.

Reviewer #2: No

Reviewer #3: **Yes: **Dr. Md. Shahgahan Miah

---

## [Author Response · Author response to Decision Letter 1]

11 Oct 2024

Response to Reviewer #3: 

Reviewer point #1: The title could be revised. ‘Adolescent girls of reproductive age’ is likely unnecessary, as adolescence represents the onset of reproductive ability, which might make the title concise. So, ‘reproductive age’ can be deleted. Additionally, the author line may be revised as 3 merging the numbers 3,4 are the same organization, ‘CARE Bangladesh’.

Author response #1: 

Thank you for catching this. We have removed these two words from the revised title. Author line is also revised. (page #1). 

Reviewer point #2: In abstract:

- The author mentioned ’12 IDIs’, ‘Two FGDs’, and ‘Two KIIs’. 12 may be write into words. Also write the age range into a bracket that may be written as ‘adolescent girls aged 15-19’, avoiding the bracket.

- KIIs with one traditional birth attendant and one local pharmacist. As the study was conducted in an urban slum, there may be chances to find a trained birth attendant along with a traditional one. Because, NGOs are implementing many maternal health projects for the slum dwellers in parallel with governments. Inclusion of trained birth attendants by interviewing may add more or new reflections and dimensions of data and strengthen the results.

- Further, ‘local pharmacist’ is not an appropriate term in the context of Bangladesh, as they are already identified as ‘drug sellers’ considering their academic background and training. See (Ahmed SM, Naher N, Hossain T, Rawal, LB. Exploring the status of retail private drug shops in Bangladesh and action points for developing an accredited drug shop model: a facility based cross-sectional study. J Pharm Policy Pract. 2017).

-Results section, started with the recommendations like ‘adolescents need professional healthcare’. It’s better to start with the maternal health information sources and a trusted one. Therefore, the existing social norms need to describe how they ‘affect’ maternal health of adolescent girls and why these social norms are being influences. Delete the word ‘impact’ as your study observed not long term results.

- In conclusion, the author should be aware of using judgmental decisions like ‘without……information, no girl or woman’ would be able to make crucial decisions…. Health. Please revise.

Author response #2: 

Thank you for your valuable suggestion. 

- We have changed all the participants’s number in numerical form. Also, ‘adolescent girls aged 15-19’ were changed from ‘adolescent girls (15-19 years old)’. (page #2) 

- Your concern with KIIs is very much valid. We agree that another KII with trained birth attendant would give better understanding over the studied topic. However, as the participant ‘traditional birth attendant’ were trained by a non-government organization, we did not search for another one. For further clarification, we are adding a small description about the traditional birth attendant in the methodology section 

- Thanks for suggesting the appropriate term. We have changed the term, accordingly, all aver the manuscript

- Result of the Abstract has been written accordingly. (page #2)

- Judgemental words were removed, accordingly

Reviewer point #3: Introduction section, the first sentence statistics were outdated. You can find the updated WHO Maternal Mortality: Key Facts data using the link https://www.who.int/news-room/fact-sheets/detail/maternal-mortality. In the line 6 of this section, the author mentioned that ‘Bangladesh has not been able to eliminate maternal mortality’, is it possible? Better to write ‘reduce’ the number following the WHO standard or like wealthier countries. However, the author summarizes the literature on maternal mortality and maternal health in global and Bangladesh contexts, health consequence and increasing risks among the adolescent. In doing so, it would be useful to further discuss the healthcare practices/facilities, especially for the adolescent group with maternal health information and behavior of Bangladesh, the role of healthcare providers, family members before the aim of the study in the introduction section. As your results already explored and you include this in your discussion part of the elder family member involvement issues in relying for health information. In the fourth para, fifth line from last, the author describe ‘women in making educated decisions…’ , could you explain it. The next line, ‘we found no studies on health information-seeking….is also judgmental. Suggesting write ‘paucity’ or ‘limited’ studies or others. In the same line the author contextualized the issue well describing slum population which citation is outdated (Khatun F. et al. 2012). The author can review the World Bank data, and even can cite Hasan MZ, Hasan AMR, Rabbani AG, Selim MA, Mahmood SS. 2022. Knowledge, attitude, and practice of Bangladeshi urban slum dwellers…… PLOS Global Public Health that identified “Around 38% (62.5 million) of the total population of this country live in urban areas and about half of them live in slums.” Please try to cite upto 2020 except classic references.

Author response #3: 

Thank you for your insightful suggestion. The statistics of first sentence were updated. About the later suggestion: we agree that we need to put updated literature. However, I think there might be some technical issues that you reviewed the track change version or else. For instance, I had changed the following sentences/words: ‘Bangladesh has not been able to eliminate maternal mortality’, ‘women in making educated decisions…’ and ‘we found no studies on health information-seeking’ in the last revised submission. Regarding your suggestions on to ‘further discuss the healthcare practices/facilities, especially for the adolescent group with maternal health information and behavior of Bangladesh, the role of healthcare providers, family members before the aim of the study in the introduction section’ is addressed accordingly. We have added a paragraph regarding this. (page # 3 & 4) 

Reviewer point #4: Methods could be better justified. This topic is likely to have significant social acceptability bias, which needs to be acknowledged. Specifically-

- The author need to contextualize the urban slum and then ‘Sattola’ describing the situation of slum, and its population in more details in the study site section.

- I believe this may be a typing error as it took ‘one month’ in 2.1 section and then ‘two months’ in the section 2.2.

- The sample size was not justified scientifically. The author could follow the data saturation principles for data collection is required here to fit in more within the qualitative paradigm. Better to include one reference here on data saturation related past/current article.

- Since the author recruited 12 married and unmarried adolescent girls, TBA and pharmacist (drug seller), it is worth mentioning the participants’ inclusion and exclusion criteria. Moreover, authors could discuss the challenges including attitudinal dimension that the team faced during data collection. Recruiting trained birth attendant and elder family member is already suggested in abstract section.

- Data collection procedures were not described well. Author could describe the research team academic background, training and experiences, interview guidelines (semi-structured or others), guidelines developing procedures, field test, interview settings (slums dwellers shared room, present about the ‘place' conducted interviews), recordings, participation type (volunteer or with pay)…. I suggest to follow or maximize the COREQ 32 items and may cited (https://www.ncbi.nlm.nih.gov/books/NBK554122/) this or any other related article.

- This is unclear what did the ‘deductive code list’ entail? As the study design is qualitative, why authors followed the deductive way is conflicting. Please revise this or explain.

- Thematic analysis procedures need to be explain more and need cited. Second paragraph of data analysis, author mentioned ‘themes were developed and recording was done…. understanding’ is unclear. Can revise following a reference. This could be more reflection on this and on the strengths (arguments, justifications for why the author use) which method brings to the analysis. In addition, ‘to confirm the accuracy…. and ‘cross-matched’ is overlap with triangulation, can revise.

- In ethical issues, the author mentioned followed …. ‘high priorities’.. need to explain how. ‘All of the participants gave their written consent’ need to revise. As the study participants were the garment workers and sat in for interview on their day off, please emphasize and highlight on the interview time, compensation, privacy and confidentiality (as they live in tiny and shared rooms) of the study participants and their shared experiences that ensured in the entire study. Because, author already described the hectic office hours of the participants and day off may their priority to get adequate rest. ‘One ethical specialist’ was recruited in the team, what was his/her roles. The author may revise the sentence … ‘destroyed data’ as the manuscript is not published yet and might need go back thorough data again.

Author response #4: 

Thanks for your valuable suggestions. 

- We have added some description about urban slums of Bangladesh. The exact population were not found from any formal sources.

- In the revised version, I edited this typing error. It would be one month, for your reference. Moreover, in the revised version, I did not put any section numbers, such as, 2.1 or 2.1

- Yes. I totally agree with your concern that the sample size was not justified scientifically. Somehow, we missed that. Thanks for your comment. We have explained that in the Sampling and Recruitment section. (page #6-7)

- Thanks for the comment on clarifying inclusion and exclusion criteria. We have incorporated your concern accordingly. (page #6-7)

- Thanks very much for this suggestion. Data collection procedures are re-written and rearranged as per your concern. The background of the data collectors were written in Sampling and Recruitment section, earlier. Procedures of guidelines developing and field test experience are described in the section Study Design. The whole methodology were revised according to COREQ. Also, we have added a supplementary file for your reference.

 - This is unclear what did the ‘deductive code list’ entail? As the study design is qualitative, why authors followed the deductive way is conflicting. Please revise this or explain.

- The concern for ‘deductive code list’ has been revised.

- Thematic analysis procedures have been cited. Other concerns have been addressed.

- We mentioned ‘high priorities’.. as it ensures to make no harm for the participant and maintain confidentiality as we promised them before interviewing. Then, there was a mistake about the consent. It would be oral consent. Information about participant’s compensation, privacy and confidentiality are added in the above sections. Explanation regarding ‘One ethical specialist’ are added. In addition, your concern are very much legit about the ‘destroyed data’. There was flaws. It was not written properly. Now, we have edited the issue. (page #9) 

Reviewer point #5: The results are interesting but not line with study aim completely. However, it requires some reorganization of the results are following-

3.1 In theme 1, the author mentioned ‘maternal health information needs for adolescent girls’ is kind of assessment but aim of study was to ‘understand the maternal health information-seeking behavior and then primary goal was assess….., the author can revise the result section following the aim of study. Like, health information sources, trusted sources, social norms that influences the behavior, influences of health information seeking behavior, barriers/challenges in seeking behavior, and then needs of health information.

3.3 Moreover, I would ask to include interview number in the quotes.

Author response #5:

Thank you for your insightful comment. 

3.1 In the revised version, I updated the foremost and specific objectives. It’s true that our major objective is to understand maternal health information seeking behaviour. However, we believe that it would be better understood if we understand the need, at first, and then, the barriers or, more specifically, the influences of social norms.(page #4) In addition, the thematic order has been revised.

3.3 Interview number in the quotes are added.

Reviewer point #6: Discussion: Suggesting knowledge is ‘accurate’ isn't very helpful with a qualitative study with such a small sample (and with no additional data, e.g. observations etc.), but instead you can reflect on how knowledge is articulated or, as you do in the paper, discussing the role of the family in supporting to provide right information. Following the current literatures in introduction and reorganize the results section, the discussion need to be improved.

Author response # 6: 

Thanks for your critical observation. We have changed such words like ‘accurate’. We have revised the discussion 

Reviewer point #7: Conclusion: The first lines in the conclusion do not give any information that was intended to be explored in the study. In conclusion, readers want to see key findings interlinked with the social norms and health information seeking behavior that are based on empirical evidence, and then link those to policy recommendations.

Author response #7: 

Thank you for your feedback. We have edited the conclusion accordingly. (page #24-25)

---

## [Decision Letter · Decision Letter 2]

20 Nov 2024

Social norms and maternal health information-seeking behavior among adolescent girls: A qualitative study in a slum of Bangladesh

PONE-D-23-34519R2

Dear Dr. Haque,

We’re pleased to inform you that your manuscript has been judged scientifically suitable for publication and will be formally accepted for publication once it meets all outstanding technical requirements.

Kind regards,

Ranjit Kumar Dehury

Academic Editor

PLOS ONE

Additional Editor Comments (optional):

Dear, authors,

The article is complying the scientific standards and accepted.

With regards,

Ranjit

Reviewers' comments:

Reviewer's Responses to Questions

**Comments to the Author**

1. If the authors have adequately addressed your comments raised in a previous round of review and you feel that this manuscript is now acceptable for publication, you may indicate that here to bypass the “Comments to the Author” section, enter your conflict of interest statement in the “Confidential to Editor” section, and submit your "Accept" recommendation.

Reviewer #3: All comments have been addressed

2. Is the manuscript technically sound, and do the data support the conclusions?

Reviewer #3: Yes

3. Has the statistical analysis been performed appropriately and rigorously? 

Reviewer #3: N/A

4. Have the authors made all data underlying the findings in their manuscript fully available?

Reviewer #3: Yes

5. Is the manuscript presented in an intelligible fashion and written in standard English?

Reviewer #3: Yes

6. Review Comments to the Author

Reviewer #3: Manuscript ID: PONE-D-23-34519R2

Manuscript Title: Social norms and maternal health information-seeking behavior among adolescent girls: A qualitative study in a slum of Bangladesh

Dear Md. Ashraful Haque,

Thank you for addressing the comments well. However, I agree that majorities of the comments addressed, but still I have a very few considerable concerns that should be considered.

1. In abstract

- The author wrote ‘12 in-depth interviews (IDIs)’ in third sentence form whereas for ‘2 focus group discussion’ and ‘key informant interviews’, wrote in capitalize form.

- The author wrote thematic analysis two times e.g. ‘furthermore, the data…..thematic analysis’. Later, ‘thematic analysis …….to analyze primary data’.

- KIIs conducted with one traditional birth attendant and one drug seller. As the study was conducted in an urban slum, there may be chances to find a trained birth attendant along with a traditional one. Effort to include trained birth attendants by interviewing may add more or new reflections and dimensions of data. Because, NGOs are implementing many maternal health projects for the slum dwellers in parallel with governments.

2. Methods

- The author presented the research team composition in the study design section. This should move into the data collection procedure section.

- Same as the second paragraph ‘we developed separate……. ‘

- In study setting, the third line ‘unprepared city’ would be unplanned.

- In sampling and recruitment section, the author wrote ‘the study used…’ but in the previous entire section, the author used to write ‘we’. Should have coherent.

- Data collection procedures, the first line should move into the study design or setting because this is the study period. The next sentence ‘the field researchers shared about themselves first’ is not clear.

7. PLOS authors have the option to publish the peer review history of their article (what does this mean?). If published, this will include your full peer review and any attached files.

Reviewer #3: No

---

## [Editor Report · Acceptance letter]

27 Nov 2024

PONE-D-23-34519R2 

PLOS ONE

Dear Dr. Haque, 

I'm pleased to inform you that your manuscript has been deemed suitable for publication in PLOS ONE. Congratulations! Your manuscript is now being handed over to our production team.

Kind regards, 

on behalf of

Dr. Ranjit Kumar Dehury 

Academic Editor

PLOS ONE